# How social capital helps communities weather the COVID-19 pandemic

**Christos A. Makridis** [1,2] *, **Cary Wu** [3] *

**1** W. P. Carey School of Business, Arizona State University, Tempe, Arizona, United States of America,
**2** Sloan School of Management, Massachusetts Institute of Technology, Cambridge, Massachusetts, United
States of America, **3** Department of Sociology, York University, Toronto, Ontario, Canada

* makridis@mit.edu (CAM); carywu@yorku.ca (CW)

## Abstract

Why have the effects of COVID-19 been so unevenly geographically distributed in the
United States? This paper investigates the role of social capital as a mediating factor for the
spread of the virus. Because social capital is associated with greater trust and relationships
within a community, it could endow individuals with a greater concern for others, thereby
leading to more hygienic practices and social distancing. Using data for over 2,700 US coun-
ties, we investigate how social capital explains the level and growth rate of infections. We
find that moving a county from the 25th to the 75th percentile of the distribution of social capi-
tal would lead to a 18% and 5.7% decline in the cumulative number of infections and deaths,
as well as suggestive evidence of a lower spread of the virus. Our results are robust to many
demographic characteristics, controls, and alternative measures of social capital.

capital helps communities weather the COVID-19
pandemic. PLoS ONE 16(1): e0245135. https://doi.
org/10.1371/journal.pone.0245135

UNITED STATES

**Data Availability Statement:** All relevant data are
within the manuscript and its Supporting
information files.

**Funding:** This research was supported by a grant
from the Canadian Institutes of Health Research
[CIHR, FRN-170368; PI: Cary Wu].

## I. Introduction

There is an increasing consensus that social capital—including trust, norms, and networks [1,
2]—may serve as one of the most important ingredients in accomplishing critical tasks in
emergency situations [3–5]. Even if physical capital is destroyed, social resilience and collabo-
ration can help communities rebound, which is especially relevant during times of national
emergencies. During outbreaks, for example, social capital can facilitate calm, peaceful, and
collective action. Experiences with recent outbreaks—SARS in 2003, the 2014 Ebola outbreak,
and Zika one year later—also suggest that outbreaks are better handled in places where there is
high social capital (see e.g., [6–12]).

The COVID-19 pandemic represents the largest world-wide shock in at least a decade, if
not a century since the 1918 Influenza. While the adverse effects of COVID-19 vary substan-
tially across countries, the ramifications of the pandemic vary at least as much domestically
across different communities. In the United States, for example, there is significant heteroge-
neity in exposure to the virus. Fig 1 shows the distribution of cumulative infections per capita
across counties as of July 27th. Whereas the median county has roughly 565 infections per
100,000 people, counties in the 90th percentile have nearly five-times as much. While there are
potentially many factors behind these cross-sectional differences, the primary purpose of this

**Competing interests:** The authors have declared that no competing interests exist.

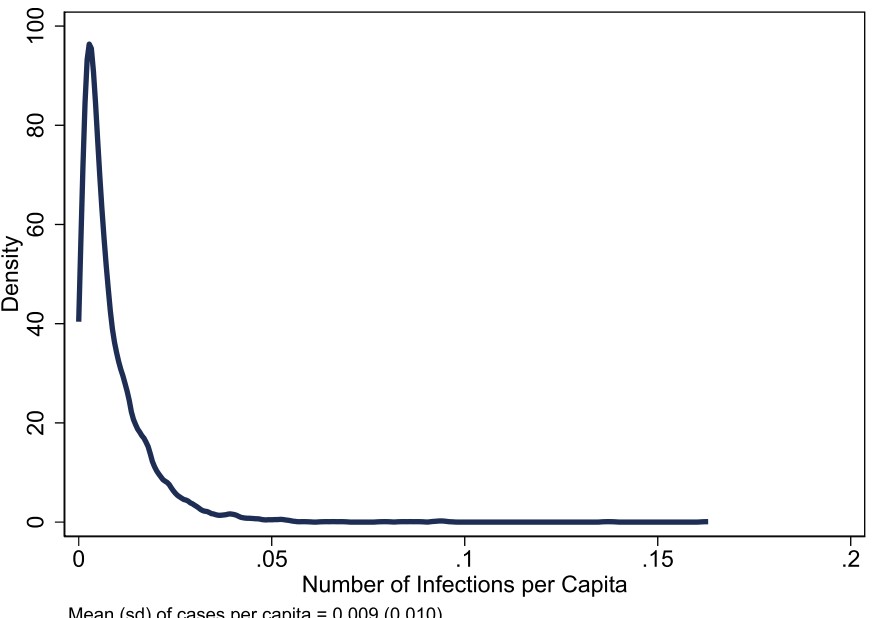

**Fig 1. Distribution of infections per capita.**

paper is to examine whether and how social capital can help communities weather the COVID-19 pandemic.

Whereas many have shown the importance of social capital during outbreaks [3, 6–8, 10, 11, 13, 14] including the on-going the COVID-19 pandemic [e.g., 15–23], theoretically, how social capital might shape the spread of the COVID-19 virus is ambiguous. In this article, we argue that because social capital is associated with greater trust and relationships within a community, it could endow individuals with a greater concern for others, thereby leading to more hygienic practices and social distancing. Using county-level data for over 2,700 US counties, we consider how differences in social capital explain the level of infections and the growth rate of infections thus far. We find that counties with a standard deviation (sd) higher level of social capital have 18% fewer infections, 5.7% fewer deaths. We also find that the rate of growth in these counties was slower.

These results are robust to controlling for a wide array of county demographic characteristics that could be correlated with the risk of infection and its spread, including population density, age and education distributions, and poverty rate. Although we recognize that these demographic controls are not exhaustive of all the omitted cross-sectional differences, we interpret the robustness of our estimates as suggestive evidence that the impact of social capital is not simply a confounding effect. We nonetheless conduct several robustness checks. For example, we find similar results when we use an alternative measure of social capital from Chetty et al. [13].

Our results are also robust to the inclusion of additional county-specific controls. Moreover, when we allow for heterogeneity in treatment effects, we show that social capital matters more in lower income counties and those with higher shares of Blacks. To understand the underlying mechanisms, we relate each of the sub-indices with COVID-19 cases, finding that family unity and collective efficacy have a significant effect above and beyond the healthcare infrastructure, which is consistent with prior evidence from Song and Lin [24]. Finally, we use Facebook's Social Connectedness Index (SCI) from Bailey et al. [25] to construct a measure of

an individuals' weighted exposure to the social capital in their social networks. We find that exposure to friends' counties with greater social capital is not associated with differences in infection and deaths after controlling for the social capital of an individual's own county. However, when we use the SCI- weighted measure of exposure to Facebook friends' social capital, we find nearly identical results.

Social capital is a multidimensional concept that consists of trust, norms, and social networks and these forms of social capital often imply different resources and indicate different aspects of social environment [26, 27]. Accordingly, they could affect social outcomes through different means. A multidimensional approach that considers how different forms of social capital affects the spread the COVID-19 will help detect the channels through which social capital makes community better able to respond to outbreaks [see also 28]. Our additional analysis shows that trust and norms indicated as family unity, community health, and collective efficacy show significant effects, while institutional health indicated using average of votes in the presidential election over 2012 and 2016, mail-back response rates for 2010 census, and confidence in institutions show no impact on COVID-19 infection. Hence, we conclude that the essential process that underlies the effect of social capital is through its association with greater trust and relationships within a community, rather ties and social networks. When individuals have a greater concern for others, they are more willing to follow hygienic practices and social distancing.

The structure of the paper is as follows. Section II provides supporting theory and background about the importance of social capital and its effects. Section III applies the theoretical framework to the COVID-19 pandemic. Section IV summarizes the data and measurement strategy. Section V outlines the empirical strategy. Section VI presents the main results. Section VII conducts a series of robustness exercises. Section VIII presents heterogeneity analyses. Section IX presents a discussion of our results in light of a growing literature. Section X concludes.

## II. Conceptualizing social capital

Social capital is intangible. While social scientists have defined it in many ways over the past few decades, there is a general recognition that it exists in social groups interactions and takes many forms. In *Making Democracy Work*, Putnam [29: 167] provides the most widely used definition, referring to social capital as "features of social organizations, such as trust, norms, and networks, that can improve the efficiency of society by facilitating coordinated actions" [29: 167]. However, we also see inconsistence across Putnam's work on the subject [see also 30]. In *Bowling Alone*, for example, Putnam [2: 19] specifically defines social capital as "connections among individuals—social networks and the norms of reciprocity and trustworthiness that arise from them". This second definition differs from the earlier one in the sense that, instead of "features" of social organizations, trust and norms become the *consequences* of social connections.

Nonetheless, across Putnam's varying definitions, three core components including degree of trust, co-operative norms as well as networks and associations are consistent [see also 30].

Following Putnam, scholars have largely come to agree that trust, norms, and networks are three essential elements of social capital [e.g., 31–33]. Trust includes not only people's faith in others but also their confidence in political institutions [34, 35]. Social norms refer to forms of social support, helping behaviors, and collective efficacy [36]. Social networks are social ties through group membership and associations that often help generate benefits or profits for individuals and social groups [37–39].

Social capital matters. It plays a key role in shaping economic [13, 40, 41] and social outcomes [2, 29, 42, 43, 39; see also 28, 44, 45]. Indeed, decades of research have shown that societies with higher levels of social capital function better [29, 38, 46], are richer [40, 47, 48], are safer [49–51], are healthier and happier [52, 53], are less corrupt [54, 55], and are more democratic [56, 57].

However, different forms of social capital often produce unequal impacts on social outcomes [e.g., 33, 58, 59]. This is because not only can different forms of social capital capture different aspects of the social environment [60], they can also imply different resources, support and obligations [26]. In fact, previous research has suggested that attitudinal social capital in forms of trust and norms is often found to be more consistent in producing positive forces than structural social capital in forms of networks or group memberships [see also 44]. While trust and civic norms are building blocks of a good and prosperous society [2, 29, 43], social ties and network relations can be used for different purposes including good, bad or neutral [38; see also 28, 44].

## III. Social capital and COVID-19

A growing body of research has suggested that, in times of crisis, higher levels of social capital may enhance individuals or communities' ability to prepare for, respond to, as well as recover from such crises [e.g., 4, 5, 53, 61, 62]. Aldrich [63], for example, finds that social capital in the form of trust among community members leads to greater sharing of information about facts, procedures or threats to the community which is critical when facing extreme events.

Individuals with few social ties are less likely to take preventative action such as evacuation, and to seek medical help or to receive assistance from others [3]. In responding to crises, trust, norms and networks enable individuals and communities easier access to various resources such as information, aid, and financial resources along with emotional and psychological support [63]. Trusting individuals are more willing to implement the necessary planning steps and share needed information and resources with others [61]. In his study of the 1995 Chicago heat wave, Klinenberg [4] shows that individuals with less social capital such as isolated, elderly, poor, and racialized individuals were not found for days after dying. Helliwell et al. [53] also show that societies with more social capital and trust can respond to economic crisis and institutional transitions more happily and more effectively.

Social capital matters not only immediately following crisis, but also afterwards during the recovery. As Aldrich writes [63: 1] writes, "social capital—the bonds which tie citizens together—functions as the main engine of long-term recovery." Lindström and Giordano [64] find that after the 2008 economic crisis social capital and trust became an important buffer against poor psychological wellbeing. Moreover, using data between 2008 and 2017, Makridis et al. [65] show that religiosity also mediates the effects of business cycle fluctuations on individual well-being. While the average individual exhibits significant cyclicality in their reported life satisfaction, active Christians not only exhibit higher levels of life satisfaction, but also acyclical levels.

Focusing specifically on epidemics and pandemics, several studies have shown that social capital is critical to the containment of outbreaks. For example, Pronyk et al. [66] and Gregson et al. [67] show that participation in local community groups is often positively associated with successful avoidance of HIV, Holtgrave et al. [68] find that more social capital is associated with lower tuberculosis case rate across US states, and Zoorob et al. [69] shows that social capital help mitigate the drug overdose epidemic in the US. Research on recent outbreaks—SARS in 2003, the 2014 Ebola outbreak, and Zika one year later—stresses the essential role that social capital and trust in particular play in preventing and controlling epidemics [see e.g., 8, 10–12].

Indeed, the World Health Organization's own outbreak guidelines [70] stress that social capital and trust are required to control and mitigate the spread of disease.

In the face of the COVID-19 outbreak, prompt isolation of those with disease, the quarantining of close contacts, and enforcement of infection control and hygiene measures are key containment measures. In fact, several studies have already looked into how social capital might shape the spread of COVID-19 [e.g., 17, 19, 21, 22, 28]. Conclusions, however, are mixed. For example, on one hand, Kuchler et al. [21] show that one dimension of social capital-social networks measured using Facebook Social Connectedness Index is strongly and positively correlated with COVID-19 prevalence across US counties. On the other hand, Wu et al. [23] use social capital and trust to explain the quality of response in the face of COVID-19 across US states. They find that states that have higher levels of social capital and trust tend to have higher testing rates.

In this paper, we argue that social capital can affect the spread of COVID-19 in two major ways. First, social capital might affect the spread of COVID-19 in the forms of its economic, health, and political benefits in pre-crisis context. Communities with more social capital often perform better economically and politically and have rich and healthy individuals [29, 40, 52]. Since wealthier communities have higher performing healthcare facilities and likely have an easier time acquiring personal protective equipment, these areas may experience a lower number of infections and a faster recovery. Moreover, wealthier individuals are concentrated in jobs that often involve less social contact and public transportation [71]. Governments may also respond faster in communities with high social capital because of greater repercussions associated with ignoring their constituents—for example, greater civic engagement to hold them accountable for inaction or bad policy [72].

Second, social capital might mitigate the spread of COVID-19 in the forms of shared norms and trust as well as networks. For example, residents in areas with greater social capital may also use more hygienic practices and greater responsibility out of trust and care for their neighbors and community members. Moreover, to contain COVID-19, governments and health officials need to rely on citizen trust in order to organize and implement effective responses. Individuals in a high social capital community have higher levels of informal guardianship, and they are more active in intervening for common good [49, 51, 73].

Lack of trust in government is associated with people's low compliance with control interventions and high refusal to adopt preventive behaviors [8, 11]. Low trust will also challenge the effective emergency responses as it will cause disruptions of community interactions, public panic, and fragmentation, and create a vicious cycle between lack of trust, non-compliance, hardships and further distrust [8, 11, 74]. Furthermore, when societies have more trust and civic norms, they become less dependent on formal institutions [45, 75]. Social capital in the forms of networks and ties facilitates collective actions, information sharing and decision-making within communities and unites people of different backgrounds and political preferences [29, 38, 72, 76, 77].

## IV. Data and measurement

To test our arguments, we need data on the spread of COVID-19 and on community characteristics. First, we measure the spread of the COVID-19 across counties using data collected by Johns Hopkins University at the county-level each day between March and July.

Fig 2 presents spatial heterogeneity in infections per capita and social capital. Counties with higher social capital cluster much more in the North, which could reflect some of the long-run and persistent effects of slavery in the south. Moreover, Fig 3 shows that there is a strong negative correlation between logged infections and social capital across counties (r = -0.24). While

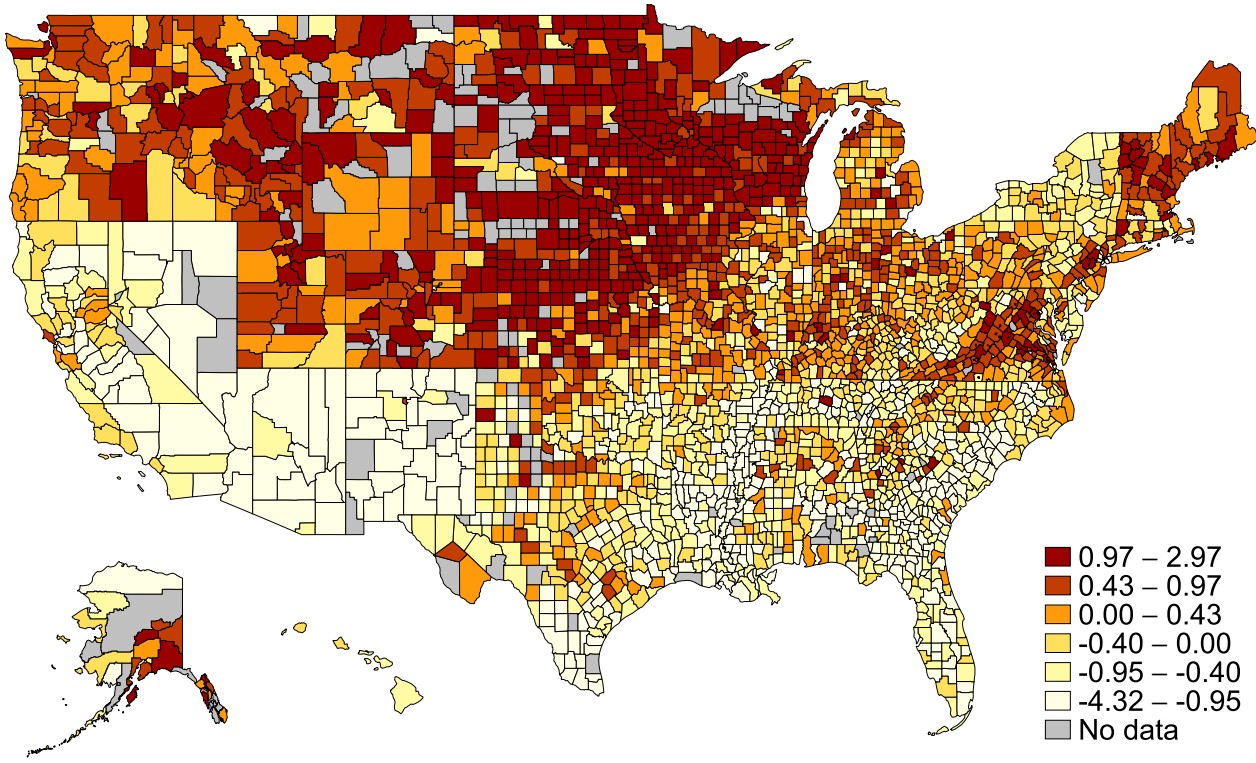

**Fig 2. Spatial heterogeneity in infections per capita and social capital.**

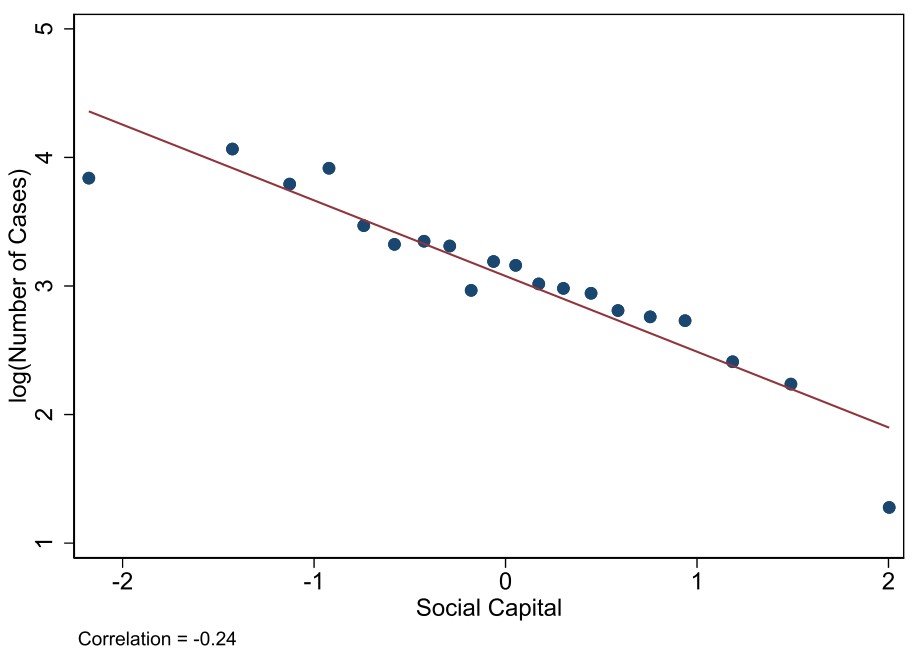

Correlation = -0.24

**Fig 3. The relationship between infections and social capital.**

the correlation is confounded by other factors, the raw data highlights the main relationship of interest.

Second, we also obtain many measures of a county's demographic characteristics between 2014 and 2018 using the Census Bureau's five-year American Community Survey (ACS). We use the Joint Economic Committee (JEC) measure of social capital at the county-level [78], which contains indicators "related to family structure and stability, family interaction and investment, civil society, trust and confidence in institutions, community cohesion, institutions, volunteerism, and social organization." These indicators, at the county-level, include: the share of births in the past year to women who were unmarried, the share of women ages 35–44 who are currently married and not separated, the share of own children living in a single-parent family, registered non-religious non- profits per 1,000, religious congregations per 1,000, an informal civil society sub-index, and the average of votes in the Presidential election per citizen ages 18 and over. The civil society sub-index includes: combination of share who volunteered, who attended a public meeting, who report having worked with neighbors to fix/improve something, who served on a committee or as an officer, who attended a meeting where politics was discussed, and who took part in a demonstration in the past year [see also 65]. These data are generally compiled based off of estimates from 2012 to 2016—well before COVID-19 and, therefore, predetermined with respect to current infection rates and the spread of the virus. Our additional analysis using the index of social capital from Chetty et al. [13] at a commuting-zone level shows similar results.

## V. Empirical strategy

To understand the relationship between COVID-19 infections and social capital, we use regressions that relate infections with the degree of social capital in a county:

$$y_{ct} = \gamma SC_c + g(X_c, \theta) + \Phi_{st} + \epsilon_{ct}$$

where $y$ denotes the logged number of (COVID-19) infections or deaths individuals within a county-day pair, $SC$ denotes a standard-normal z-score of social capital in the county, $g(X, \theta)$ denotes a semi-parametric function of demographic controls, and $\Phi$ denotes state-by-day fixed effects. We cluster standard errors at the county-level to allow for geographic autocorrelation.

We include a wide array of controls within our semi-parametric function, $g(X, \theta)$, including: the age distribution (share of individuals under age 18, age 18–24, age 25–34, 35–64, and 65+), the education distribution (the share of individuals 25 and older with less than a high school degree, a high school degree, some college, and more than college), the racial distribution (the share of individuals who are white and the share who are black), the share of married households, the poverty rate (for individuals below age 18, between 18–64, and 65+), and population density. We control for the unemployment rate and real gross domestic product (GDP) growth in 2012 prices using data recently made available by the Bureau of Economic Analysis (BEA).

Our inclusion of these demographic characteristics and population density is especially important since one potential threat to identification is that areas with higher social capital are more rural and less likely to experience infection due to less social contact. Moreover, our inclusion of state fixed effects allows us to compare counties within the same state. This is potentially important since states have exerted varying degrees of emergency powers, ranging from strict stay-at-home orders to more standard quarantines. These differences in state policy could be correlated with social capital if communities vary in their tastes and elect different

governors to power. While there are many potential concerns about omitted variables, we explore these possibilities in Section VII.

## VI. Main results

Table 1 presents the main results associated with regressions of logged number of cases, logged number of deaths, and the week-to-week case growth in the number of cases averaged within each month on social capital, conditional on different layers of controls and/or fixed effects. The raw data suggests that a standard deviation (sd) increase in social capital is associated with a 59.2% decline in the number of cases, a 32.4% decline in the number of deaths, and a 5 percentage point (pp) decline in the weekly growth in the number of cases (columns 1, 4, and 7). However, an important concern with these cross-sectional results is the presence of omitted variables: for example, counties with higher social capital have lower population density and, therefore, are less likely to experience as much of the pandemic than other areas, like New York City.

We now include a wide array of semi-parametric controls for a county's demography, including population density and the age distribution. Not surprisingly, we see that a 1% rise in population density is associated with a 0.43% rise in the number of cases and a 0.44pp rise in the weekly growth in cases. We also control for the age and education distributions, together with the share of males and married households and the poverty rate for different age brackets.

Consistent with the concern that omitted factors lead to an overestimation over the role of social capital, since these counties vary in other positive ways, the magnitudes of the estimated coefficients decline, but remain statistically significant. For example, we now find that a 1sd rise in social capital is associated with a 17.8% decline in the number of cases, a 5.7% decline in the number of deaths, and a 3pp decline in the growth rate of cases (columns 2, 5, and 8). Only the growth rate is not statistically significant at the 10% level.

We further address concerns about the presence of omitted variables bias and cross-sectional differences across states by including state-by-day fixed effects, which allows us to exploit variation across counties in the same state on a given day. If, for example, states with higher levels of social capital lead to the voting in of certain politicians that pass better policy that contains the virus, then we might be concerned that our results are simply driven by different political choices. This also controls for heterogeneity in state social distancing policies that could also be correlated with social capital. Although our estimated effect when the outcome variable is the logged number of deaths becomes statistically insignificant, our main result in column 3 remains statistically significant.

Although we have controlled for day-of-the-year fixed effects in our most restrictive specification, one concern is that these results are driven by differences in infection rates in one part of the pandemic over another. We explore the time series behavior of social capital as a mediating force by regressing logged cases on social capital separately for each day, plotting the estimated coefficient with its corresponding confidence interval in Fig 4. Although the point estimate oscillates between -0.20 and -0.45 between April and July, it is always negative, consistent with the view that social capital dampens the spread of the virus, particularly during waves of the spread.

## VII. Robustness exercises

We first test the sensitivity of our results to alternative measures of social capital, recognizing that the definition remains contested. We separately regress the logged number of cases on z-scores of the four different inputs that go into the overall score from the Joint Economic Committee. We also include another measure of social capital from Chetty et al. [13], which has

**Table 1. The effects of social capital on infections and the spread of COVID-19.**

| Dep. var. = | log(Number of Cases) | | | log(Number of Deaths) | | | Weekly Growth | |
|---|---|---|---|---|---|---|---|---|
| | **(1)** | **(2)** | **(3)** | **(4)** | **(5)** | **(6)** | **(7)** | **(8)** |
| Social capital (z-score) | -.592*** | -.178*** | -.125*** | -.324*** | -.057** | .029 | -.050*** | -.030 |
| | [.029] | [.031] | [.038] | [.020] | [.023] | [.027] | [.013] | [.019] |
| log(Population Density) | | .702*** | .744*** | | .430*** | .453*** | | .095*** |
| | | [.031] | [.041] | | [.020] | [.026] | | [.013] |
| Age Under 18, % | | 5.588*** | 5.228*** | | 4.848*** | 2.113 | | 3.692*** |
| | | [1.693] | [1.572] | | [1.399] | [1.315] | | [1.091] |
| Age 18–24, % | | 1.241 | 1.932 | | -1.327 | -1.606 | | .493 |
| | | [1.547] | [1.411] | | [1.319] | [1.177] | | [.965] |
| Age 35–64, % | | 1.699 | 1.915 | | .940 | .589 | | .115 |
| | | [1.518] | [1.405] | | [1.240] | [1.094] | | [.924] |
| Age 65+, % | | 1.356 | -.343 | | 3.478*** | 1.490 | | 2.202*** |
| | | [1.305] | [1.211] | | [1.081] | [.992] | | [.839] |
| White, % | | -1.026** | -.630 | | -1.369*** | -1.309*** | | -.189 |
| | | [.400] | [.506] | | [.272] | [.310] | | [.182] |
| Black, % | | .244 | .526 | | -.269 | .150 | | -.008 |
| | | [.383] | [.435] | | [.270] | [.290] | | [.191] |
| Less than High School, % | | 3.224*** | 3.056*** | | .662* | 2.024*** | | -.141 |
| | | [.519] | [.639] | | [.370] | [.442] | | [.337] |
| Some College, % | | 1.362*** | -.279 | | -.095 | .011 | | -.347 |
| | | [.426] | [.582] | | [.338] | [.436] | | [.279] |
| College, % | | 5.415*** | 3.026*** | | 3.052*** | 2.270*** | | 1.389*** |
| | | [.676] | [.815] | | [.558] | [.624] | | [.422] |
| Post-graduate, % | | 2.561** | 2.254 | | 2.866*** | 2.071** | | -.174 |
| | | [1.151] | [1.452] | | [.936] | [1.004] | | [.583] |
| Male, % | | .561 | -.519 | | .921 | -1.220* | | 1.246* |
| | | [1.045] | [.951] | | [.766] | [.722] | | [.662] |
| Married, % | | -3.313*** | -1.552** | | -2.568*** | -1.142*** | | -1.352*** |
| | | [.519] | [.615] | | [.388] | [.408] | | [.351] |
| Poverty rate under 18, % | | .098 | .376 | | .047 | .161 | | -.281 |
| | | [.369] | [.346] | | [.257] | [.243] | | [.239] |
| Poverty rate 18–64, % | | -4.050*** | -3.195*** | | -2.546*** | -1.708*** | | -1.023** |
| | | [.580] | [.550] | | [.416] | [.390] | | [.398] |
| Poverty rate 65+, % | | -.558 | -1.314* | | .950* | .356 | | .417 |
| | | [.794] | [.754] | | [.517] | [.501] | | [.360] |
| R-squared | .06 | .35 | .82 | .05 | .38 | .65 | .00 | .02 |
| Sample Size | 437166 | 437017 | 436868 | 437166 | 437017 | 436868 | 249298 | 249210 |
| Controls | No | Yes | Yes | No | Yes | Yes | No | Yes |
| State x Day FE | No | No | Yes | No | No | Yes | No | No |

*Notes.*–Sources: Joint Economic Committee, Census Bureau, and Johns Hopkins COVID-19 Tracker, March—July 2020. The table reports the coefficients associated with regressions of daily county logged number of cases, logged number of deaths, and the week-to-week growth rate averaged within each month of the number of cases on a standardized *x*-score of county social capital, conditional on county demographic controls, which include: logged population density, the age distribution (normalized to the share of individuals between ages 25 and 34), the education distribution (normalized to the share of individuals with a high school degree), the share of males, and the share of married households. Standard errors are clustered at the county-level.

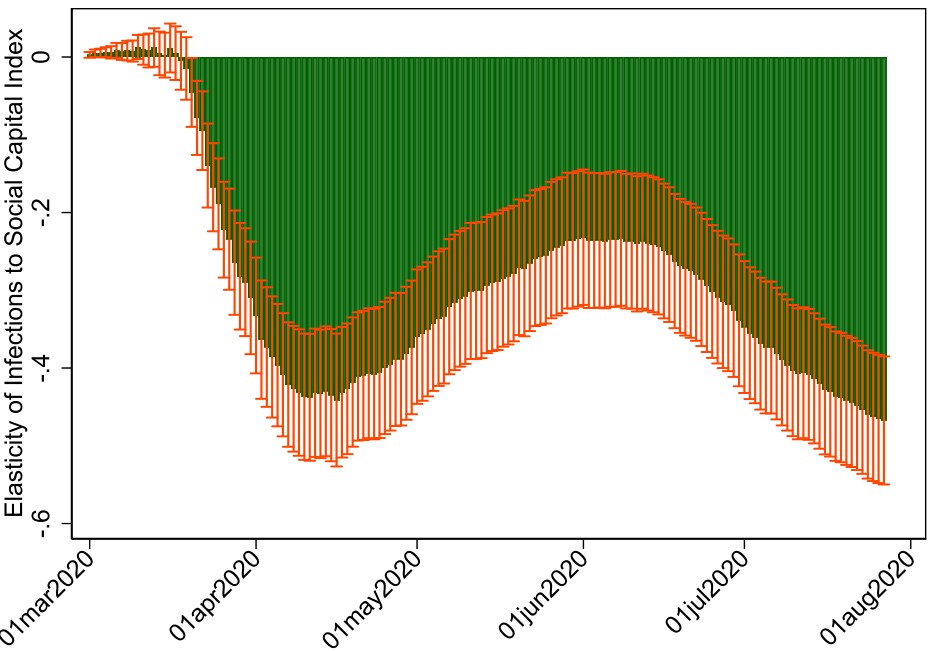

**Fig 4. Elasticity of cumulative infections to social capital, March-July 2020.**

0.60 correlation with the JEC measure. In this sense, while there is no perfect measure of social capital, these results are comforting in that they point towards similar conclusions. These results are summarized in Table 2.

We begin with our baseline results in columns 1 and 2, which show the cross-sectional relationship and conditional correlations. While family unit is negatively correlated with the number of infections in the cross-section, the correlation disappears after introducing our controls. This does not necessarily imply that family structure is irrelevant for the number of infections, but rather that most of the association likely operates through selection effects. Moreover, the measure of family structure from the JEC is based on a limited and coarse set of indicators. For example, when we use marital status as a proxy for family structure, we find a robust and positive association with both the level of infections and its weekly growth rate even conditional on our standard controls.

Not surprisingly, our measure of community health is highly predictive of infections. Institutional health is as well, but interestingly the cross-sectional correlation implies a positive association even though the conditional correlation flips negative, as in our main result. One possible explanation may stem from the fact that areas with greater institutional health are likely more urban and wealthy areas, but those areas have lower social capital. Finally, our baseline effects are nearly the same when we use the measure from Chetty et al. [13].

An additional concern with our results thus far is that differences in social capital could be correlated with direct measures of economic activity. Using 2018 data on county-level real GDP (in 2012 prices), we now include the growth rate of GDP between 2017 and 2018. Since there is some persistence in real GDP, we interpret this growth rate as a reasonable proxy for differences, although we find similar results when we use the average year-to-year growth between 2010 and 2018, for example. Including the county GDP growth rate does not alter our estimate on social capital at all, although we find that a percentage point rise in county GDP growth from 2017 to 2018 is associated with a 0.46% decline in the number of cases, which

**Table 2. Examining evidence of heterogeneous treatment effects.**

| Dep. var. = | log(Number of Cases) | | | | Weekly Growth | | | |
|---|---|---|---|---|---|---|---|---|
| | (1) | (2) | (3) | (4) | (5) | (6) | (7) | (8) |
| Social capital | -.283*** | -.244*** | -.269*** | -.246*** | .024 | .049** | .079*** | .076*** |
| | [.032] | [.036] | [.035] | [.038] | [.018] | [.023] | [.021] | [.023] |
| 1[Income > Median] | | .213*** | | | | -.029 | | |
| | | [.046] | | | | [.030] | | |
| × Social capital | | -.088** [.044] | | | | -.046* [.027] | | |
| 1[Density > Median] | | | .194*** | | | | -.140*** | |
| | | | [.059] | | | | [.030] | |
| × Social capital | | | -.009 [.043] | | | | -.126*** [.023] | |
| 1[Black Share > Median] | | | | .197*** | | | | .062** |
| | | | | [.044] | | | | [.026] |
| × Social capital | | | | -.046 [.054] | | | | -.088*** [.027] |
| R-squared | .76 | .76 | .76 | .76 | .52 | .52 | .53 | .52 |
| Sample Size | 355096 | 355096 | 355096 | 355096 | 253800 | 253800 | 253800 | 253800 |
| Controls | Yes | Yes | Yes | Yes | Yes | Yes | Yes | Yes |
| Day FE | Yes | Yes | Yes | Yes | Yes | Yes | Yes | Yes |

*Notes.*–Sources: Joint Economic Committee, Census Bureau, and Johns Hopkins COVID-19 Tracker, March—July 2020. The table reports the coefficients associated with regressions of daily county logged number of cases and the average county week-to-week growth rate in number of cases on a standardized $x$-score of county social capital, an indicator for whether the county ranks above the median with respect to a particular demographic variable, and their interaction, conditional on county demographic controls, which include: logged population density, the age distribution (normalized to the share of individuals between ages 25 and 34), the education distribution (normalized to the share of individuals with a high school degree), the share of males, and the share of married households. The median household income ("income") across counties is $51,259; median population density ("density") across counties is 63; the median share of African Americans ("AA Share") across counties is 3.5%. Standard errors are clustered at the county-level.

reflects the fact that areas experiencing greater growth have more resources to deal with the crisis.

Another concern is that our outcome variable may contain measurement error in the number of true infections or the number who have yet to be identified. While measurement in the outcome variable will attenuate our estimated effect [79], we nonetheless address this concern by introducing the logged number of total test results and the ratio of positive to negative test results as additional controls. Unfortunately, these are only available at the state-level, but examining how our main coefficient changes after introducing the control is useful. Including both these variables as controls does not produce statistically different results. Moreover, since we are controlling for state-by-day fixed effects, our most restrictive specification already purges differences in state-level testing rates that could affect the responsiveness of a given county in the state.

To further test that it is not through social networks and institutional trust that social capital affects Americans' COVID-19 response [19], we draw on data from Facebook's Social Connectedness Index (SCI), which was introduced by Bailey et al. [25]. The SCI measures the number of friendship ties between one county, denoted $c$, and every other county in the United States, denoted $c'$. We normalize the number of friendship ties for each county by dividing by the total number of friendship ties. Then, we construct an SCI-weighted social capital index as follows:

$$SC_c^{SCI} = \sum_{c'} SC_c' SCI_{c,c'}$$

number of friendship ties between county $c$ and $c'$. Using this new measure of SCI-weighted social capital, we regress the number of logged cases on not only county social capital, but also the SCI-weighted social capital index. Importantly, we remove county $c$ from its contribution to $SC^{SCI}$ so that there is no mechanical correlation between our two right-hand-side variables.

Although we find a coefficient of -0.183 (p-value = 0.022) on SCI-weighted social capital without any demographic controls, and a coefficient of -0.496 (p-value = 0.00) on local social capital, the correlation on SCI-weighted social capital declines substantially to -0.048 (p-value = 0.418) after adding controls. Moreover, the coefficient on local social capital is -0.156 (p-value = 0.00), which is similar to our baseline specification presented earlier. This suggests that, while social networks play a role in influencing individuals' adherence to social distancing measures, it cannot account for the effect of local social capital on infections, suggesting that we have a genuine effect.

While we can use SCI-weighted social capital as a control, and are comforted by the fact that it is not statistically associated with infections or death after controlling for a county's own social capital, the strong correlation between the two of 0.80 suggests that it could serve as an instrument. Indeed, when we instrument for a county's own social capital using its exposure to other counties, we find almost identical results: a 1sd rise in social capital is associated with a 21% decline in cumulative cases, although the standard errors are slightly larger (p-value < 0.01). When we use logged cumulative deaths as the outcome variable, we find a coefficient of -0.05, but it is statistically insignificant at conventional levels (p-value = 0.210). Moreover, when we use logged cases as the outcome variable, but include state-by-day fixed effects, we find a coefficient of -0.24, although the p-value is 0.171. Our F-statistic under this specification is 91, suggesting that it passes the relevance test. This builds on related research that social networks can contribute to differences in health by propagating the types of social relationships that different members of a group engage in [80]

Our identifying assumption is that differences in a county's exposure to friends' social capital affects the spread of the virus in their own county only through the effects on their own social capital. While one potential violation is that counties with higher social capital are simply connected to other counties with higher social capital, thereby leading to the same selection effects, the instrument directly addresses the potential concern that counties with higher social capital have lower infections because of better healthcare infrastructures. Instead, these results are consistent with the interpretation that greater social capital leads to greater collective efficacy in times of crisis.

## VIII. Heterogeneity analysis

We finally turn towards dimensions of heterogeneity. Table 2 documents the results associated with regressions of the logged number of cases and the growth in weekly infections on social capital, together with an interaction for whether the county ranks above the median (across counties) with respect to a particular demographic variable, namely: median household income, population density, and the share of Black households. We are specifically motivated by evidence that minorities and lower income communities have been more adversely affected, so we want to explore whether social capital continues to play a mediating role even in these lower income communities.

Columns 1 and 5 report the baseline effects with added day-of-the-year fixed effects to control for common macroeconomic shocks. Turning towards the three types of heterogeneity, we indeed find that the association between social capital and the pandemic is concentrated among higher income counties, more dense counties, and even counties with higher shares of African Americans. For example, a 1sd rise in social capital is associated with a 33% (4.6pp)

decline in the number of cases (growth) in counties ranking above the median of household income and an additional 24% decline (4.9pp increase) for counties ranking below the median (columns 2 and 7), although the coefficient for weekly growth as the outcome variable is statistically insignificant.

While we find no statistically significant differences between infections and social capital when we split above and below the median of population density and the share of Blacks—the direct effects are all negative, just as in the baseline—we find some differences when our outcome is the weekly growth in infections averaged within a month. For example, social capital behaves as more of a mediating force in counties that rank above the median in population density and above the median in the share of Blacks, which is important since these are precisely the sets of counties that experienced greater risk of infections and the spread of infections.

Finally, given the historical evidence of religion as an important determinant of social capital [2, 81, 82], together with evidence that religion helps individuals weather crises during economic hardship [65], we also examine religiosity as a potential mediating factor. We use data from the Association of Statisticians of American Religious Bodies to measure religious adherence at a county-level. Although it has flaws, the 2010 U.S. Religion Census remains the best available data source for membership and adherence data at the county-level. We create an indicator for whether the county ranks above the median with respect to the share of religious adherents, interacting the indicator with our standardized index of social capital. In the raw data, we find that social capital behaves as a complement: high adherence counties have an additional 0.24% decline in cumulative cases for a 1sd rise in social capital. However, the coefficient declines in magnitude to 0.033% when we control for our demographic characteristics.

## IX. Discussion

While current studies, including ours here, have found a similar negative relationship between social capital and intensity of the COVID-19 virus [15, 17–22], the underlying mechanisms that dictate how social capital helps cushion locations over the pandemic remain less clear. Most recently, Wu [28] argues that, because different forms of social capital have different implications and can yield distinct influences, studying how they affect the COVID-19 response can help detect the specific mechanisms underlying.

For example, analyzing a survey data from China, Wu [23] shows that trust in government shows the strongest effect in reducing an individual's exposure to COVID-19, while social trust has little impact. In an authoritarian country, like China, social capital affects the response largely through individuals' compliance with top-down control measures, rather bottom-up collective actions. Like our study here, Ding et al. [19] have also considered how social capital might help explain why U.S. counties respond differently to COVID-19. They separate between two forms of social capital—community engagement and individual commitment to social institutions—finding that these two forms of social capital produce opposite impacts on social distancing. On one hand, people in counties where individuals historically engaged less in community activities are more likely to practice social distancing. On the other hand, they also find that people in counties where individuals have greater willingness to commit to societal institutions and norms are more likely to engage with social distancing. In the U.S., social capital helps compared to communities with lower social capital, communities with higher levels of social capital are better able to respond to the COVID-19 pandemic through Americans' greater commitment to achieving a common goal.

To understand the mechanisms at play, we partitioned our primary index of social capital into its relative components and showed heterogeneous effects (see Table 3). We found that

**Table 3. Robustness results with alternative social capital measures.**

| Dep. var. = | log(Number of Cases) | | | | | | | | | | | |
|---|---|---|---|---|---|---|---|---|---|---|---|---|
| | (1) | (2) | (3) | (4) | (5) | (6) | (7) | (8) | (9) | (10) | temp11 | temp12 |
| Social capital | -.592*** | -.178*** | | | | | | | | | | |
| | [.029] | [.031] | | | | | | | | | | |
| Family unity | | | -.290*** | -.073** | | | | | | | | |
| | | | [.031] | [.031] | | | | | | | | |
| Community health | | | | | -.957*** | -.204*** | | | | | | |
| | | | | | [.029] | [.030] | | | | | | |
| Institutional health | | | | | | | .031 | -.018 | | | | |
| | | | | | | | [.031] | [.024] | | | | |
| Collective efficacy | | | | | | | | | -.605*** | -.085*** | | |
| | | | | | | | | | [.045] | [.023] | | |
| Chetty et al. [13] | | | | | | | | | | | -.513*** | -.099*** |
| | | | | | | | | | | | [.026] | [.027] |
| R-squared | .06 | .35 | .01 | .35 | .15 | .37 | .00 | .37 | .06 | .36 | .04 | .37 |
| Sample Size | 437166 | 437017 | 441040 | 440891 | 458324 | 458175 | 454599 | 454450 | 441934 | 441785 | 454897 | 454748 |
| Controls | No | Yes | No | Yes | No | Yes | No | Yes | No | Yes | No | Yes |
| State FE | No | No | No | No | No | No | No | No | No | No | No | No |
| Day FE | No | No | No | No | No | No | No | No | No | No | No | No |

*Notes.*–Sources: Joint Economic Committee, Census Bureau, and Johns Hopkins COVID-19 Tracker, March—July 2020. The table reports the coefficients associated with regressions of daily county logged number of cases and the average county week-to-week growth rate in number of cases on a standardized *x*-score of overall county social capital from the JEC, its four separate inputs, and a measure of social capital from Chetty et al. [13], conditional on county demographic controls, which include: logged population density, the age distribution (normalized to the share of individuals between ages 25 and 34), the education distribution (normalized to the share of individuals with a high school degree), the share of males, and the share of married households. Standard errors are clustered at the county-level.

trust and norms indicated as family unity, community health, and collective efficacy show significant effects. However, institutional health indicated using average of votes in the presidential election over 2012 and 2016, mail-back response rates for 2010 census, and confidence in institutions show no impact on COVID-19 infection. This suggests that, in the case of the U.S., social capital affects response to COVID-19 through trust and norms, rather social networks and institutional trust. In this sense, the underlying mechanism behind the relationship between social capital and infections operate through greater trust and relationships within a community. When individuals have a greater concern for others, they are more willing to follow hygienic practices and social distancing.

## X. Conclusion

The world has experienced a steady increase in the frequency of disease outbreaks over the past three decades. Given the growing rate of travel, urbanization, and interaction with the environment and natural resources [83], these trends may continue even as medicine advances. Understanding the role that social forces and institutions play in mediating the effects of public health crises is important for disciplining public health interventions and retaining stability and resilience throughout a crisis so that countries do not plummet into civil strife.

This paper provides the first account of the effects of social capital on COVID-19 infections and the underlying mechanisms that explain these effects. Rather than leading to a spread of the virus through greater social interaction, social capital has a significant negative effect on

the number of infections and growth of the virus. For example, if we moved a county that ranks in the 25[th] percentile to the 75[th] percentile in the distribution of social capital, our base-line estimates suggest that their infections would decline by roughly 18%. Moreover, the virus would not have spread as fast. These results are not driven by omitted variables, such as population density or the age distribution, and hold among comparisons of counties in the same state. In sum, trust and social cohesion are integral for managing a crisis [4, 5].

Our results also have clear policy implications: public health interventions cannot be dis-connected from the social forces that are present at a local level. Moreover, stable and vibrant communities are not luxuries, but rather important priorities for managing emergencies. Investing in social capital and interpersonal relationships helps us manage negative shocks and retain levels of interconnectedness and well-being. Continued research is required to under-stand the evolution of social capital and the specific mechanisms through which it affects human flourishing during crises.

## Supporting information

**S1 File.**
(ZIP)

## Author Contributions

**Conceptualization:** Christos A. Makridis, Cary Wu.

**Data curation:** Christos A. Makridis, Cary Wu.

**Formal analysis:** Christos A. Makridis.

**Methodology:** Christos A. Makridis.

**Validation:** Christos A. Makridis.

**Visualization:** Christos A. Makridis.

**Writing – original draft:** Christos A. Makridis.

**Writing – review & editing:** Christos A. Makridis, Cary Wu.

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
