## [Decision Letter · Decision Letter 0]

13 Nov 2020

PONE-D-20-26570

Ties That Bind (and Social Distance): How Social Capital Helps Communities Weather the COVID-19 Pandemic

PLOS ONE

Dear Dr. Makridis,

Thank you for submitting your manuscript to PLOS ONE. After careful consideration, we feel that it has merit but does not fully meet PLOS ONE’s publication criteria as it currently stands. Therefore, we invite you to submit a revised version of the manuscript that addresses the points raised during the review process.

This is an interesting paper that addresses an important topic. Two reviewers have completed thoughtful reviews. Prior to publication, please clarify and revise as necessary the spatial statistics used in the paper, and please clarify the concept of a social networks.

We look forward to receiving your revised manuscript.

Kind regards,

Jacob Freeman

Academic Editor

PLOS ONE

Journal Requirements:

2. We note that Figure 2 in your submission contains map images which may be copyrighted. All PLOS content is published under the Creative Commons Attribution License (CC BY 4.0), which means that the manuscript, images, and Supporting Information files will be freely available online, and any third party is permitted to access, download, copy, distribute, and use these materials in any way, even commercially, with proper attribution. For these reasons, we cannot publish previously copyrighted maps or satellite images created using proprietary data, such as Google software (Google Maps, Street View, and Earth). For more information, see our copyright guidelines: http://journals.plos.org/plosone/s/licenses-and-copyright.

(1) You may seek permission from the original copyright holder of Figure 2 to publish the content specifically under the CC BY 4.0 license. 

3. Please include a copy of Table 4 which you refer to in your text on pages 19 and 20.

4. We note you have included a table to which you do not refer in the text of your manuscript. Please ensure that you refer to Table 1 in your text; if accepted, production will need this reference to link the reader to the Table.

Reviewers' comments:

Reviewer's Responses to Questions

**Comments to the Author**

1. Is the manuscript technically sound, and do the data support the conclusions?

Reviewer #1: Partly

Reviewer #2: Yes

2. Has the statistical analysis been performed appropriately and rigorously? 

Reviewer #1: Yes

Reviewer #2: Yes

3. Have the authors made all data underlying the findings in their manuscript fully available?

Reviewer #1: Yes

Reviewer #2: Yes

4. Is the manuscript presented in an intelligible fashion and written in standard English?

Reviewer #1: Yes

Reviewer #2: Yes

5. Review Comments to the Author

Reviewer #1: Comment(s) on the criteria I used to evaluate the paper:

- I applaud the authors for the thoroughness of their analysis. They did an impressive array of tests and largely persuaded me that they were truly identifying the influence of social capital on variation across US counties in covid_19 infection rates. I have one primary remaining concern. Namely, I do not think the authors properly deal with the problem of spatial correlation, which is critical in this type of analysis. And, state fixed effects nor clustering at the county level properly deal with this concern. For instance, New York City was perhaps the worst hit place on earth during the period of their study (March thru July). This was almost certainly due to a number of factors such as the early arrival of the virus relative to most other counties in the US (due to high rates of international and domestic visitors) and high population density (i.e., factors such as high usage of public transportation). The authors models control for this for the counties of NYC. Yet, their models do not control for a county’s proximity to other hard-hit spots. Thus, regardless of levels of social capital (as well as income, etc.), nearby counties in New Jersey, Connecticut, (and therefore possibly Northeast PA, RI, etc.) will certainly be more likely to struggle to contain the virus due to proximity to hard-hit areas (which proxies for interactions across counties with areas with high infection rates). Thus, I would like to see some kind of spatial analysis done, such as Conley standard errors (for perhaps the nearest 100 miles) or other means for controlling for proximity of counties to hard-hit (and not hard-hit) areas.

Other comments which I did not use in evaluating this manuscript:

- I have not attempted to evaluate whether other authors have conducted and/or published work that is substantially similar to this work. I’m not familiar enough with this rapidly expanding literature. So, I cannot say how original a contribution this analysis is.

- I think Section 2, especially sub-section B, is too long. In my opinion, this could be substantially condensed. The authors should get to their analysis more quickly.

- While the paper is well written, it includes some typos. For some reason, the authors did not include page numbers; so, I’m listing the errors by PDF pages, as provided by the journal.

PDF p. 10 (top) – both of these sentences include typos – “….in a county is exposed to through social networks. We find that exposure to friends’ in counties with greater social capital is not associated with differences in infection…”

PDF p. 14 (last line) – “…..were not be found for days after dying.”

Reviewer #2: This well-written manuscript presents evidence that social capital mitigates the spread of COVID-19. The authors examine both variables—as well as a set of control variables—at the county level, using regressions. Results are significant, and this paper could contribute to our understanding of how social processes mediate public health outcomes. My main comment is that the paper could benefit from greater clarification of the role of social networks (both in terms of measurement, as well as conceptualization of the relationship between networks and other indicators of social capital).

Major point:

1) Consistent with much of the literature on social capital, the authors include social networks as a component/dimension of social capital. However, the measure of social networks used—Facebook’s Social Connectedness Index (SCI)—is an imperfect indicator of social networks, for two reasons. First, the SCI only accounts for relationships on Facebook. More importantly, the SCI focuses on relationships with individuals from other counties. While one can assume that these between-county relationships indicate bridging social capital, they do not emphasize the more proximate relationships that relate more to bonding social capital. Additionally, social networks are strongly related to other indicators of social capital (e.g., community cohesion, family structure). Indeed, the authors note that “the underlying mechanism behind the relationship between social capital and infections operate through greater trust and relationships within a community” (page 27) – social networks encompass relationships within a community. I encourage the authors to clarify their use of the term “social networks” throughout the paper to explain that the network variable only measures one component of social networks. Likewise, the authors may consider explaining how social interaction is interdependent with certain other indicators of social capital.

Minor points:

1) The abstract notes that social capital could contribute to infection rates (“On one hand, higher social capital could imply greater in-person interaction and risk of contagion”; this possibility is mentioned again on page 9). However, this idea is not developed (e.g., as an alternate hypothesis). Its emphasis early in the manuscript may lead readers to expect a more prominent role in the manuscript. The authors may consider either dropping this idea from the abstract and Introduction, or developing it further (e.g., in section iii and/or section ix).

2) First paragraph of the Introduction: “Even if capital is destroyed…” – it is not clear what “capital” refers to (other forms of social capital?).

6. PLOS authors have the option to publish the peer review history of their article (what does this mean?). If published, this will include your full peer review and any attached files.

Reviewer #1: No

Reviewer #2: No

---

## [Editor Report · Decision Letter 1]

17 Dec 2020

PONE-D-20-26570R1

Ties That Bind (and Social Distance): How Social Capital Helps Communities Weather the COVID-19 Pandemic

PLOS ONE

Dear Dr. Makridis,

Thank you for submitting your manuscript to PLOS ONE. After careful consideration, we feel that it has merit but does not fully meet PLOS ONE’s publication criteria as it currently stands. Therefore, we invite you to submit a revised version of the manuscript that addresses the points raised during the review process.

ACADEMIC EDITOR:

Please proofread the article carefully as we do not do a formal copy edit. In addition, I have made several comments below that require minor revision. Thank you for submitting your work to Plos One.

1) p. 3 ``Although we recognize that these demographic

controls are not exhaustive of all the omitted cross-sectional differences, we interpret the robustness

of our estimates as suggestive evidence that additional bias would not qualitatively change our

results. We nonetheless report conduct several robustness checks.”

The word choice of bias is odd here. Do you mean additional variables. Also, remove `report’ or ‘conduct’ from the last sentence.

2) ``from Bailey et al. (2018) to construct a measure of the social capital that

an individual in a county is exposed to social networks.

Something is not right here. Not clear what you mean be construct a measure of social capital that an individual in a county is exposed to social networks.

3) ``Helliwell et al.

(2014) also shows that societies with more social capital and trust can respond to economic crisis

and institutional transitions more happily and more effectively.”

Check this; I think `shows’ should be `show’ If the subject is the authors’ article rather than the article, then it should be plural. Please check throughout the manuscript.

4) ``Furthermore, when societies have more trust and civic norms, they

become less depend on formal institutions (Arrow 1972; Knowles 2007).”

dependent instead of depend?

5)``We also conduct

robustness using the index of social capital from Chetty et al. (2014) at a commuting zone-level.”

The robustness of what to what? Your results to changes in how you estimate social capital? Please clarify.

6) Black and African American seem to be used interchangeably throughout. Pick one.

7) ``In the U.S., social capital helps communities better able to respond to

the COVID-19 pandemic through Americans’ greater commitment to achieving a common goal.”

`better able to’ should be something like ``higher social capital helps communities respond better than communities with lower social capital” the use of `better’ requires a comparison.

8) Figure 2. Please use an A and B to clarify which map displays social capital and infection data. Please clarify in the caption.

We look forward to receiving your revised manuscript.

Kind regards,

Jacob Freeman

Academic Editor

PLOS ONE

---

## [Author Response · Author response to Decision Letter 1]

22 Dec 2020

See the "Response to Reviewers.docx"

---

## [Editor Report · Decision Letter 2]

23 Dec 2020

How Social Capital Helps Communities Weather the COVID-19 Pandemic

PONE-D-20-26570R2

Dear Dr. Makridis,

We’re pleased to inform you that your manuscript has been judged scientifically suitable for publication and will be formally accepted for publication once it meets all outstanding technical requirements.

Kind regards,

Jacob Freeman

Academic Editor

PLOS ONE
---

## [Editor Report · Acceptance letter]

30 Dec 2020

PONE-D-20-26570R2 

How Social Capital Helps Communities Weather the COVID-19 Pandemic 

Dear Dr. Makridis:

I'm pleased to inform you that your manuscript has been deemed suitable for publication in PLOS ONE. Congratulations! Your manuscript is now with our production department. 

Kind regards, 

on behalf of

Dr. Jacob Freeman 

Academic Editor

PLOS ONE